# Robust Outlier-Adaptive Filtering for Vision-Aided Inertial Navigation

**DOI:** 10.3390/s20072036

**Published:** 2020-04-04

**Authors:** Kyuman Lee, Eric N. Johnson

**Affiliations:** 1School of Aerospace Engineering, Georgia Institute of Technology, 270 Ferst Drive, Atlanta, GA 30313, USA; 2Faculty of Aerospace Engineering, The Pennsylvania State University, 229 Hammond Building, University Park, PA 16802, USA; eric.johnson@psu.edu

**Keywords:** V-INS, UAV, EKF, IMU, camera vision, computer vision, image processing, outlier rejection, adaptive filtering, sensor fusion, navigation

## Abstract

With the advent of unmanned aerial vehicles (UAVs), a major area of interest in the research field of UAVs has been vision-aided inertial navigation systems (V-INS). In the front-end of V-INS, image processing extracts information about the surrounding environment and determines features or points of interest. With the extracted vision data and inertial measurement unit (IMU) dead reckoning, the most widely used algorithm for estimating vehicle and feature states in the back-end of V-INS is an extended Kalman filter (EKF). An important assumption of the EKF is Gaussian white noise. In fact, measurement outliers that arise in various realistic conditions are often non-Gaussian. A lack of compensation for unknown noise parameters often leads to a serious impact on the reliability and robustness of these navigation systems. To compensate for uncertainties of the outliers, we require modified versions of the estimator or the incorporation of other techniques into the filter. The main purpose of this paper is to develop accurate and robust V-INS for UAVs, in particular, those for situations pertaining to such unknown outliers. Feature correspondence in image processing front-end rejects vision outliers, and then a statistic test in filtering back-end detects the remaining outliers of the vision data. For frequent outliers occurrence, variational approximation for Bayesian inference derives a way to compute the optimal noise precision matrices of the measurement outliers. The overall process of outlier removal and adaptation is referred to here as “outlier-adaptive filtering”. Even though almost all approaches of V-INS remove outliers by some method, few researchers have treated outlier adaptation in V-INS in much detail. Here, results from flight datasets validate the improved accuracy of V-INS employing the proposed outlier-adaptive filtering framework.

## 1. Introduction

The most widely used algorithms for estimating the states of a dynamic system are a Kalman Filter [1,2] and its nonlinear versions such as an extended Kalman filter (EKF) [3,4]. After the NASA Ames Research Center implemented the Kalman filter into navigation computers to estimate the trajectory of the Apollo program, engineers have developed a myriad of applications of the Kalman filter in navigation system research areas [5]. For example, Magree and Johnson [6] developed a simultaneous localization and mapping (SLAM) architecture with improved numerical stability based on the UD factored EKF, and Song et al. [7] proposed a new EKF system that loosely fuses both absolute state measurements and relative state measurements. Furthermore, Mostafa et al. [8] integrated radar odometry and visual odometry via EKF to help overcome their limitations in navigation. Despite the development of numerous applications of the Kalman filter in various fields, it suffers from inaccurate estimation when required assumptions fail.

Estimation using a Kalman filter is optimal when process and measurement noise are Gaussian. However, sensor measurements are often corrupted by unmodeled non-Gaussian or heavy-tailed noise. An abnormal value relative to an overall pattern of the nominal Gaussian noise distribution is called an outlier. In other words, in statistics, an outlier is an observation that deviates so much from other observations as to arouse suspicion that it is generated by a different mechanism [9]. Such outliers have many anomalous causes. They arise due to unanticipated changes in system behavior (e.g., temporary sensor failure or transient environmental disturbance) or unmodeled factors (e.g., human errors or unknown characteristics of intrinsic noise). As an example of measurement outliers in many navigation systems, either computer vision data contaminated by outliers or sonar data corrupted by phase noise lead to erroneous measurements. Process outliers also occur by chance. Inertial measurement unit (IMU) dead reckoning and wheel odometry as a proxy often generate inaccurate dynamic models in visual-inertial odometry (VIO) and SLAM algorithms, respectively. Without accounting for outliers, the accuracy of the estimator significantly degrades, and control systems that rely on high-quality estimation can diverge.

### 1.1. Related Work

#### 1.1.1. State Estimation for Measurements with Outliers

As the performance of the Kalman filter degrades at the presence of measurement outliers, many researchers have investigated other approaches to mitigate the impact of the outliers. Mehra [10] created adaptive filtering with the identification of noise covariance matrices and showed the asymptotic convergence of the estimates towards their true values. Maybeck [11] and Stengel [12] found other noise-adaptive filtering such as covariance matching. However, all of these filters performed only offline and required filter tuning. To estimate parameter values in unknown covariances without the need for manual parameter tuning, Ting et al. [13] used a variational expectation–maximization (EM) framework. That is, they introduced a scalar weight for each observed data sample and modeled the weights to be Gamma distributed random variables. However, it assumed that noise characteristics are homogeneous across all measurements even though sensors have distinct properties. Särkkä and Nummenmaa [14] provided the online learning of the parameters of the measurement noise variance, but to simultaneously track the system states and the levels of sensor noise, they additionally defined a heuristic transition model for the noise parameters. Piché et al. [15] developed Gaussian assumed density filtering and smoothing framework for nonlinear systems using the multivariate Student *t*-distribution, and Roth et al. [16] included an approximation step for heavy-tailed process noise, but these filters are not applicable in high dimensions. Next, Solin and Särkkä [17] found that the added flexibility of Student-*t* processes over Gaussian processes robustifies inference in outlier-contaminated noisy data, but they treated only analytic solutions enabled by the noise entanglement.

Recently, Agamennoni et al. developed the outlier-robust Kalman filter (ORKF) [18,19] to obtain the optimal precision matrices of measurement outliers by variational approximation for Bayesian inference [20]. However, this method requires iterations at every time, even when observed data contain no outliers. Graham et al. also established the $l_{1}$-norm filter [21] for both types of sparse outliers. However, the filter might not work for nonlinear systems since they derived the constraint of $l_{1}$-norm optimization based on only linear system equations. Similar to the ORKF, the $l_{1}$-norm filter needs the constrained optimization at all times, even when no additional noise present as outliers. Therefore, these two approaches demand some extensive computational complexity for either iterations or optimization. As outliers do not always arise (i.e., are rare), we reduce such computation cost if a test detects the time when outliers occur. All of the above methods were not validated for complicated systems such as unmanned aerial vehicles or vision-aided inertial navigation [6,22,23] or with sequential measurement updates [24,25].

#### 1.1.2. Outlier Rejection Techniques

One of the primary problems in vision-aided inertial navigation systems (V-INS) is incorrect vision data correspondence or association. Matched features between two different camera views are corrupted by outliers because of image noise, occlusions, and illumination changes that are not modeled by the feature matching techniques. To provide cleaned measurement data to the filter, outlier removal in image processing front-end is essential. One of standard outlier rejection techniques is RANdom SAmple Consensus (RANSAC) [26]. RANSAC is an iterative approach to estimate the parameters of a mathematical model from a set of observed data contaminated by outliers. An underlying assumption is that the data consist of inliers whose distribution is described by some set of the parameters of the model and outliers that do not fit the model. The generated parameters are then verified on the remaining subset of the data, and the model with the highest consensus is a selected solution. In particular, 2-point RANSAC [27,28] is an extended RANSAC-based method for two consecutive views of a camera rigidly mounted on a vehicle platform. Given gyroscopic data from IMU measurements, randomly selected two-feature correspondences hypothesize an ego-motion of the vehicle. This motion constraint discards wrong data associations in the feature matching processes.

For detecting remaining outliers that are not rejected in the image processing front-end, outlier detection tests are required in filtering back-end. Most of the statistical tests [29] that require access to the entire set of data samples for detecting outliers might not be a viable option in real-time applications. For example, the typicality and eccentricity data analysis [30,31] used in [32] is an inadequate measure in V-INS, as computing the means and the variances of each residual of sequential measurements is challenging. In V-INS, the tracking of some measured features is possibly lost due to out of sight, and new feature measurements are coming for initialization.

For the real-time outlier detection of sequential measurements in V-INS, the Mahalanobis gating test [33] is a useful measure based on the analysis of residual and covariance signals at each feature measurement. The approach builds upon each Mahalanobis distance [34] of residuals and compares each value against a threshold given by the quantile of the chi-squared distribution with a certain degree of freedom. The confidence level of the threshold is designated prior to examining the data. Most commonly, the 95% confidence level is used. This hypothesis testing, called goodness of fit, is a commonly used outlier detection method in practice. Because of such suitability of the Mahalanobis gating test to real-time detection in V-INS, this paper combines the test with the ORKF [18,32] to detect and handle measurement outliers in vision-aided estimation problems. Similar to the derivation of update steps for handling measurement outliers in the ORKF, for computing the optimal precision matrices of unmodeled outliers in V-INS, Section 4 will derive feasible update procedures by variational inference. In other words, whenever unexpected outliers appear, the outlier-adaptive filtering in this paper updates and marginalizes measurement outliers to improve the robustness of the navigation systems.

### 1.2. Summary of Contributions

This paper presents improving the use of outlier removal techniques in image processing front-end and the development of a robust and adaptive state estimation framework for V-INS when frequent outliers occur. For outlier removal in the image processing front-end of V-INS, feature correspondence constitutes the following three steps: tracking, stereo matching, and 2-point RANSAC. To estimate the states of V-INS in which vision measurements still contain remaining outliers, we propose a novel approach that combines a real-time outlier detection technique with an extended version of the ORKF in the filtering back-end of V-INS. Therefore, our approach does not restrict noise at either a constant or Gaussian level in filtering. The testing results of benchmark flight datasets show that our approach leads to greater improvement in accuracy and robustness under severe illumination environments.

Starting from the architecture of the existing vision-aided inertial navigation system, this paper more focuses on contributing to the development of red boxes in Figure 3.

### 1.3. A Guide to This Document

The remainder of this document contains the following sections. Section 2 introduces background for all of this study. To estimate the states of V-INS in which frequent outliers arise, Section 3 examines outlier rejection techniques in image processing front-end, and Section 4 and Section 5 formulate a novel implementation of robust outlier-adaptive filtering. Section 6 shows testing results of this study on benchmark flight datasets. The last section concludes and plans future work.

## 2. Preliminaries

### 2.1. The Extended Kalman Filter

The system equations with continuous-time dynamics and a discrete-time sensor are as follows,
(1)$$\begin{array}{cc} \dot{x}\left(t\right) & =f\left(\;x(t),\;\eta (t)\;\right) \end{array}$$
(2)$$\begin{array}{cc} y(t_{k}) & =h\left(\;x\left(t_{k}\right)\;\right)+\zeta \left(t_{k}\right), \end{array}$$
where $x\in R^{n}$ is the state and $y\in R^{m}$ a measurement. f(·) and h(·) are the nonlinear dynamic and measurement functions, respectively. Let us assume that these functions are known based on each equation of motion and modeling. To clarify, *t* denotes continuous time, subscript *k* represents the *k*-th time step, and initial condition $x\left(0\right)=x_{0}$ is given. Moreover, let us assume that both propagation and measurements are corrupted by additive zero-mean white Gaussian noise; that is, η(t)∼N(0,Q) and $\zeta \left(t_{k}\right)∼N\left(0,R\right)$.

#### 2.1.1. Time Update

To estimate the state variables of the system, we design a hybrid EKF in the following steps. In the propagation step, state estimate $\hat{x}:=E\left[x\right]$ and its error-covariance $P:=E[\;\left(x-\hat{x}\right){\left(x-\hat{x}\right)}^{T}\;]$ are integrated from time ${\left(k-1\right)}^{+}$ to time $k^{-}$ with respect to variable τ
$${\hat{x}}_{k}^{-}={\hat{x}}_{k-1}^{+}+\int_{t_{k-1}}^{t_{k}}f\left(\hat{x}\left(\tau \right)\right)\;d\tau ,$$
where let ${\hat{x}}_{k}=\hat{x}\left(t_{k}\right)$. Hat “ ^ ” denotes an estimate, and superscripts − and + represent a priori and a posteriori estimates, respectively. Here, for one numerical solution of the ordinary differential equation, we use the Heun’s method [35] that refers to the improved Euler’s method or a similar two-stage Runge–Kutta method. Jacobian *A*, *B*, and state transition matrix Φ are defined by
(3)$$\begin{array}{cc} A_{k-1} & =\frac{\partial f(x)}{\partial x}{|}_{{\hat{x}}_{k-1}^{+}},\;\;\;\;B_{k-1}=\frac{\partial f(x)}{\partial \eta }{|}_{{\hat{x}}_{k-1}^{+}}\\ \Phi_{k-1} & =\exp\left(A_{k-1}\;\Delta t_{k-1}\right)\approx I+A_{k-1}\;\Delta t_{k-1}, \end{array}$$
where $\Delta t_{k-1}=t_{k}-t_{k-1}$. Letting $P_{k}=P\left(t_{k}\right)$, the time update of error covariance is
(4)$$P_{k}^{-}=\Phi_{k-1}\;P_{k-1}^{+}\;\Phi_{k-1}^{T}+B_{k-1}\;Q\;B_{k-1}^{T}\;\Delta t_{k-1}.$$

#### 2.1.2. Measurement Update

Using actual sensor measurements, the measurement update step of the EKF corrects state estimates and its corresponding error covariance after propagation. Letting $y_{k}=y\left(t_{k}\right)$, at every time *k* when $y_{k}$ is measured,
(5)$$\begin{array}{cc} K_{k} & =P_{k}^{-}C_{k}^{T}\;{\left(\;C_{k}P_{k}^{-}C_{k}^{T}+R\;\right)}^{-1} \end{array}$$
(6)$$\begin{array}{cc} {\hat{x}}_{k}^{+} & ={\hat{x}}_{k}^{-}+K_{k}\;\left(\;y_{k}-h\left({\hat{x}}_{k}^{-}\right)\;\right) \end{array}$$
(7)$$\begin{array}{cc} P_{k}^{+} & =P_{k}^{-}-K_{k}C_{k}P_{k}^{-}, \end{array}$$
where *K* is called the Kalman gain and Jacobian *C* is defined as
$$C_{k}=\frac{\partial h(x)}{\partial x}{|}_{{\hat{x}}_{k}^{-}}.$$

Equation (Equation 7) is the Joseph’s form [36] of the covariance measurement update, so this form preserves its symmetry and positive definite. For more details such as optimality and derivation, see References [24,37].

#### 2.1.3. Sequential Measurement Update

When myriad measurements are observed at one time, sequential Kalman filtering is useful. In fact, we obtain *N* measurements, $y_{1},y_{2},\cdots ,y_{N}$, at time *k*; that is, we first measure $y_{1}$, then $y_{2}$, ⋯, and finally $y_{N}$, shown in Figure 1.

We first initialize a posteriori estimate and covariance after zero measurement is processed; that is, they are equal to the a priori estimate and covariance. For i=1,⋯,N, perform the general measurement update using the *i*-th measurement. We lastly assign the a posteriori estimate and covariance as ${\hat{x}}_{k}^{+}\leftarrow {\left({\hat{x}}_{k}\right)}_{N}$ and $P_{k}^{+}\leftarrow {\left(P_{k}\right)}_{N}$. Based on Simon [24]’s proof that the sequential Kalman filtering is equivalent formulation of the standard EKF, the order of updates does not affect overall performance of estimation.

### 2.2. Models of Vision-Aided Inertial Navigation

#### 2.2.1. Vehicle Model

The nonlinear dynamics of a vehicle is driven by raw micro-electromechanical system (MEMS) IMU sensor data including specific force and angular velocity inputs. The estimated vehicle state is given by
(8)$${\hat{x}}_{V}={\left[\begin{array}{ccccc} {}^{i}{\hat{p}}_{b/i}^{T} & {}^{i}{\hat{v}}_{b/i}^{T} & {\delta \hat{\theta }}^{T} & {\hat{b}}_{a}^{T} & {\hat{b}}_{\omega }^{T} \end{array}\right]}^{T},$$
where $p_{b/i}$, $v_{b/i}$ are the position and velocity of the vehicle with respect to the inertial frame, respectively. δθ is the error quaternion of the attitude of the vehicle, and its more details are explained in [38,39,40]. $b_{a}$ and $b_{\omega }$ are the acceleration and gyroscope biases of the IMU, respectively. Left superscript *i* denotes a vector expressed in the inertial frame. The EKF propagates the vehicle state vector by dead reckoning with data from the IMU. Raw MEMS IMU sensor measurements $a_{\mathrm{raw}}$ and $\omega_{\mathrm{raw}}$ are corrupted by noise and bias as follows,
$$\begin{array}{cc} a_{\mathrm{raw}} & =a_{\mathrm{true}}-T_{b/i}\;{}^{i}g+b_{a}+\eta_{a}\\ \omega_{\mathrm{raw}} & =\omega_{\mathrm{true}}+b_{\omega }+\eta_{\omega } \end{array}$$
(9)$$\begin{array}{cc} {\dot{b}}_{a} & =\eta_{b_{a}} \end{array}$$
(10)$$\begin{array}{cc} {\dot{b}}_{\omega } & =\eta_{b_{\omega }}, \end{array}$$
where $a_{\mathrm{true}}$ and $\omega_{\mathrm{true}}$ are the true acceleration and angular rate, respectively, and *g* is the gravitational acceleration in the inertial frame. $\eta_{a}$ and $\eta_{\omega }$ are zero-mean, white, Gaussian noise of the accelerometer and gyroscope measurement, and $T_{b/i}={T_{i/b}}^{T}$ denotes the rotation matrix from the inertial frame to the body frame.

$\eta_{b_{a}}$ and $\eta_{b_{\omega }}$ in Equations (Equation 9) and (Equation 10) are the random walk rate of the acceleration and gyroscope biases. From the works in [40,41,42], the MEMS accelerometer and gyroscope are subject to flicker noise in the electronics and other components susceptible to random flickering. The flicker noise causes their biases to wander over time. Such bias fluctuations are usually modeled as a random walk. In other words, slow variations in the bias of the IMU sensor are modeled with a “Brownian motion” process, also termed a random walk in discrete-time. In practice, the biases are constrained to be within some range, and thus the random walk model is only a good approximation to the true process for short periods of time.

The vehicle dynamics is given by
(11)$$\begin{array}{cc} {}^{i}{\dot{\hat{p}}}_{b/i} & ={}^{i}{\hat{v}}_{b/i} \end{array}$$
(12)$$\begin{array}{cc} {}^{i}{\dot{\hat{v}}}_{b/i} & ={\hat{T}}_{i/b}\;\left(\;a_{\mathrm{raw}}-{\hat{b}}_{a}\;\right)+{}^{i}g \end{array}$$
(13)$$\begin{array}{cc} {\dot{\hat{q}}}_{i/b^{\prime }} & =\frac{1}{2}\;Q\left(\;\omega_{\mathrm{raw}}-{\hat{b}}_{\omega }\;\right)\;{\hat{q}}_{i/b^{\prime }} \end{array}$$
(14)$$\begin{array}{cc} \delta \dot{\hat{\theta }} & =-\left\lceil \;(\;\omega_{\mathrm{raw}}-{\hat{b}}_{\omega }\;)\;\times \;\right\rfloor \delta \hat{\theta } \end{array}$$
(15)$$\begin{array}{cc} {\dot{\hat{b}}}_{a} & =0 \end{array}$$
(16)$$\begin{array}{cc} {\dot{\hat{b}}}_{\omega } & =0, \end{array}$$
where $\left\lceil \alpha \;\times \right\rfloor$ is a skew symmetric matrix, and function Q(·) maps a 3 by 1 vector of the angular velocity into a 4 by 4 matrix [25]. The use of the 4 by 1 quaternion representation in state estimation causes the covariance matrix to become singular, so it requires considerable accounting for the quaternion constraints. To avoid these difficulties, engineers developed the error-state Kalman filter in which 3 by 1 infinitesimal error quaternion δθ is used instead of 4 by 1 quaternion *q* in the state vector. In other words, we use attitude error quaternion to express the incremental difference between tracked reference body frame $b^{\prime }$ and actual body frame *b* for the vehicle. Jacobian matrix $A=\frac{\partial \;\dot{x}}{\partial \;x}{|}_{\hat{x}}$ and $B=\frac{\partial \;\dot{x}}{\partial \;\eta },\;\mathrm{where}\;\eta ={\left[\;\eta_{a}^{T},\eta_{\omega }^{T},\eta_{b_{a}}^{T},\eta_{b_{\omega }}^{T}\;\right]}^{T},$ are computed in Appendix A.

#### 2.2.2. Camera Model

An intrinsically calibrated pinhole camera model [27,43] is given by
(17)$$\begin{array}{cc} \left[\begin{array}{c} u_{j}\\ v_{j} \end{array}\right]=y_{j} & =h_{j}\left(x\right)+\zeta_{j}=\left[\begin{array}{c} f_{u}\frac{{}^{c}X_{j}}{{}^{c}Z_{j}}\;+\zeta_{u_{j}}\\ f_{v}\frac{{}^{c}Y_{j}}{{}^{c}Z_{j}}\;+\zeta_{v_{j}} \end{array}\right] \end{array}$$
(18)$$\begin{array}{cc} {\left[\begin{array}{ccc} {}^{c}X_{j}, & {}^{c}Y_{j}, & {}^{c}Z_{j} \end{array}\right]}^{T}={}^{c}p_{f_{j}/c} & =T_{c/i}\;\left(\;{}^{i}p_{f_{j}/i}-{}^{i}p_{c/i}\;\right)\\ & =T_{c/b}\;T\left(q_{i/b}\right)\;\left(\;{}^{i}p_{f_{j}/i}-{}^{i}p_{b/i}\;\right)-T_{c/b}\;{}^{b}p_{c/b}, \end{array}$$
where *x* is the state vector including the vehicle state and the feature state, and measurement $y_{j}$ is the *j*-th feature 2*D* location on the image plane. $f_{u}$ and $f_{v}$ are the horizontal and vertical focal lengths, respectively, and $\zeta_{u}$ and $\zeta_{v}$ are additive, zero-mean, white, Gaussian noise of the measurement. Vectors $p_{f_{j}/c}$, $p_{f_{j}/i}$ are the *j*-th feature 3*D* position with respect to the camera frame and the inertial frame, respectively. Extrinsic parameter $T_{c/b}$ and ${}^{b}p_{c/b}$ are known and constant. Jacobian matrix $C_{j}=\frac{\partial \;y_{j}}{\partial \;x}{|}_{\hat{x}}$ is computed in Appendix A. In addition, from Equation (Equation 19), if *j*-th measurement $y_{j}$ on an image is a new feature, then ${}^{i}p_{f_{j}/i}$ is unknown so need to be initialized. Details of feature initialization are explained in Appendix B.

## 3. Outlier Rejection in Image Processing Front-End

### 3.1. Feature Correspondence

In this study, a feature detector using the Features from Accelerated Segment Test (FAST) algorithm [44,45] maintains a minimum number of features in each image. For each new image, a feature extractor using the Kanade–Lucas–Tomasi (KLT) sparse optical flow algorithm [46] tracks the existing features. Even though Paul et al. [47] proved that descriptor-based methods for temporal feature tracking are more accurate than KLT-based methods, as Sun et al. [48] found that descriptor-based methods require much more computing resource with a small gain in accuracy, we employ the KLT optical flow algorithm in the image processing front-end of this study. Next, our stereo matching using a fixed baseline stereo configuration also applies to the KLT optical flow algorithm for saving computational loads compared to other stereo matching approaches. With the matched features, a 2-point RANSAC [26] is applied to remove remaining outliers by utilizing the RANSAC step in the fundamental matrix test [27]. In the scope of this study, we implement the 2-point RANSAC algorithm by simply running one of open source codes.

Similar to [48,49], our outlier rejection is composed of three steps, shown in Figure 2. We assume that features from previous $c_{1}$ and $c_{2}$ images are outlier-rejected points, where $c_{1}$ and $c_{2}$ are left and right camera frames of a stereo camera, respectively. The three steps form a closed loop of previous and current frames of left and right cameras. The first step is the stereo matching of tracked features on current $c_{1}$ image to $c_{2}$ image. The next steps are applying 2-point RANSAC between previous and current images of the left camera and another 2-point RANSAC between previous and current images of the right camera. For steps 2 and 3, stereo-matched features are directly used in each RANSAC.

### 3.2. Algorithm of Feature Correspondence

Algorithm 1 summarizes the feature correspondence for outlier rejection. For the scope of this paper, the OpenCV library [50] and open source codes of RANSAC are extremely useful and directly applied.
**Algorithm 1** Feature Correspondence for Outlier Rejection
Require: Pyramids and outlier-rejected points of previous $c_{1}$, $c_{2}$ images
1:*Feature Tracking*: 2:**function**buildOpticalFlowPyramid(current $c_{1}$ or $c_{2}$ image)                           ▹ OpenCV [50]3: **return** pyramid of current $c_{1}$ or $c_{2}$
4:**end function**5:**function**predictFeatures(outlier-rejected points of previous $c_{1}$, ${\hat{T}}_{\mathrm{curr}\leftarrow \mathrm{prev}}$ of $c_{1}$, Intrinsic $c_{1}$) 6: **return** predicted features of current $c_{1}$
7:**end function**8:**function**calcOpticalFlowPyrLK(pyramids of previous and current $c_{1}$, outlier-rejected points of previous $c_{1}$, predicted features of current $c_{1}$)                                            ▹ OpenCV [50]9: **return** tracked points of previous $c_{1}$ and $c_{2}$, tracked features of current $c_{1}$
10:**end function**11:*Stereo Matching*: 12:**function**stereoMatching(tracked points of previous $c_{1}$ and $c_{2}$, tracked features of current $c_{1}$) 13: Initialize $c_{2}$ points by projecting the tracked features of current $c_{1}$ to $c_{2}$ using the rotation from stereo extrinsic14: **function**
calcOpticalFlowPyrLK(pyramid of current $c_{1}$ and $c_{2}$, tracked features of current $c_{1}$, initialized $c_{2}$ points)                                                    ▹ OpenCV [50]15: **end function**16: Further remove outliers based on the essential matrix17: **return** matched points of previous $c_{1}$ and $c_{2}$, matched features of current $c_{1}$ and $c_{2}$
18:**end function**19:*2-Point RANSAC*:20:**function**twoPointRansac(matched points of previous $c_{1}$ or $c_{2}$, matched features of current $c_{1}$ or $c_{2}$, ${\hat{T}}_{\mathrm{curr}\leftarrow \mathrm{prev}}$ of $c_{1}$ or $c_{2}$, Intrinsic of $c_{1}$ or $c_{2}$) 21: **return** outlier-rejected points of current $c_{1}$ or $c_{2}$
22:**end function**23:*Addition of Newly Detected Features*: 24:Create a mask to avoid re-detecting existing features 25:**function**fastFeatureDetector(current $c_{1}$ image, mask)26: **return** new features on current $c_{1}$
27:**end function**28:**function**stereoMatching(new features on current $c_{1}$) 29: **return** matched new features on current $c_{2}$
30:**end function**31:Group all of outlier-rejected features

In Algorithm 1, Pyramid is a type of multi-scale signal representation in which an image is subject to repeated smoothing and sub-sampling.

## 4. Outlier Adaptation in Filtering Back-End

Even though image processing front-end removes outliers by tracking, stereo matching, and 2-point RANSAC, some outlier features still survive and enter the filter as inputs. This section explains the outlier rejection procedure in filtering back-end.

### 4.1. Outlier Removal in Feature Initialization

If a measurement is a new feature, our system initializes its 3*D* position with respect to the inertial frame. In feature initialization, Gaussian–Newton least-squares minimization first estimates the depth of left $c_{1}$ camera. If either the estimated depth of the left or right camera is negative, then the solution of the minimization is invalid since features are always in front of both camera frames observing them. This process of removing features that has the invalid depth is referred to as outlier removal in feature initialization.

### 4.2. Outlier Detection by Chi-Squared Statistical Test

Before operating navigation systems, we initialize the chi-squared test table with a 95% confidence level. While the systems estimate the state variable, if *j*-th measurement $y_{j}$ at time *k* is the existing feature, then its residual $r_{j}$ and Jacobian $C_{j}$ are computed. Next, we proceed a Mahalanobis gating test [33] for residual $r_{j}$ to detect remaining outliers. In fact, Mahalanobis distance [34] $\gamma_{j}$ is a measure of the distance between residual $r_{j}$ and covariance matrix $S_{j}=C_{j}\;{\left(P_{k}\right)}_{j-1}\;C_{j}^{T}+R$
(19)$$\gamma_{j}=r_{j}^{T}\;{\left(C_{j}\;{\left(P_{k}\right)}_{j-1}\;C_{j}^{T}+R\right)}^{-1}\;r_{j}.$$In the statistic test, we compare $\gamma_{j}$ value against a threshold given by the 95-th percentile of the $\chi^{2}$ distribution with $\nu_{j}$ degree of freedom. Here, $\nu_{j}$ is the number of observations of the *j*-th feature minus one. If the feature passes the test, the EKF uses residual $r_{j}$ to process the measurement update.

### 4.3. Outlier-Adaptive Filtering

Unlike the extended ORKF (EORKF) [32], for a practical estimation approach in V-INS, this study investigates only measurement outliers due to the following reasons. As the measurement update is not the process performed at every time step, the outlier detection by each residual value cannot directly detect the outliers of IMU measurements. Furthermore, in the sequential measurement update, multiple residuals are computed to update at one IMU time stamp. In other words, as only rare observations among feature measurements from one image are corrupted by the remaining outliers, hypothesizing that the outliers come from the IMU may be faulty. Therefore, in the scope of this paper, we handle only measurement outliers.

#### 4.3.1. Student’s *t*-Distribution

Despite the true system with outliers, the classical EKF assumes that each model in the filter is corrupted with additive white Gaussian noise. The levels of the noise are assumed to be constant and encoded by sensor covariance matrices *Q* and *R* (i.e., $\eta_{k}∼N\left(0,\;Q\right),\;{\left(\zeta_{j}\right)}_{k}∼N\left(0,\;R\right)$). However, as outliers arise in the realistic system, now we do not restrict noise at either a constant or Gaussian level. Instead, their levels vary over time, or noise have heavier tails than the normal distribution as follows,
(20)$$\zeta_{j}{|}_{k}∼\mathrm{ST}\left(0,\;{\tilde{R}}_{j},\;\nu_{j}\right),\;\;\;\;\;\mathrm{where}\;\;\;\;{\tilde{R}}_{j}∼W^{-1}\left(\;\nu_{j}\Lambda_{j},\;\nu_{j}\;\right),$$
where ST(·) denotes a Student’s *t*-distribution, and $\nu_{k}>m-1$ is degrees of freedom. Covariance matrix ${\tilde{R}}_{j}$ follows the inverse-Wishart distribution, denoted as $W^{-1}\left(\cdot \right)$. $\Lambda_{j}≻0$ is m×m precision matrix.

In Bayesian statistics, the inverse-Wishart distribution is used as the conjugate prior for the covariance matrix of a multivariate normal distribution [20]. The probability density function (pdf) of the inverse-Wishart is
(21)$$\begin{array}{cc} p({\tilde{R}}_{j}\;|\;\nu_{j},\Lambda_{j}) & \propto |{\tilde{R}}_{j}{|}^{-\frac{\nu_{j}+m+1}{2}}\exp\left[-\frac{\nu_{j}}{2}\;\mathrm{tr}\left(\Lambda_{j}\;{\tilde{R}}_{j}^{-1}\right)\right], \end{array}$$
where tr(·) denotes the trace of a square matrix in linear algebra. Moreover, in probability and statistics, a Student’s *t*-distribution is any member of a family of continuous probability distributions that arises when estimating the mean of a normally distributed population in situations where the standard deviation of the population is unknown [51]. Whereas a normal distribution describes a full population, a *t*-distribution describes samples drawn from a full population; thus, the larger the sample, the more the distribution resembles a normal distribution. Indeed, as the degree of freedom goes to infinity, the *t*-distribution approaches the standard normal distribution. In other words, when the variance of a normally distributed random variable is unknown and a conjugate prior placed over it that follows an inverse-Wishart distribution, the resulting marginal distribution of the variable follows a Student’s *t*-distribution [52]. Then, the Student-*t*, a sub-exponential distribution with much heavier tails than the Gaussian, is more prone to producing outlying values that fall far from its mean.

#### 4.3.2. Variational Inference

The purpose of filtering is generally to find the approximations of posterior distributions $p(x_{k}\;|\;y_{1:k})$, where $y_{1:k}=\left[y_{1},y_{2},\cdots ,y_{k}\right]$ is the histories of sensor measurements obtained up to time *k*. For systems with the heavy-tailed noise, we also wish to produce another inference about covariance matrices whose priors follow the inverse-Wishart distribution. Hence, our goal in this section is to find both approximations for posterior distribution $p(x_{1:k},{\tilde{R}}_{1:k}|y_{1:k})$ and model evidence $p(y_{1:k})$. Compared to sampling methods, the variational Bayesian method performs approximate posterior inference at low computational cost for a wide range of models [20,52]. In the method, we decompose log marginal probability
(22)$$\ln p\left(y_{1:k}\right)=\mathrm{KL}\left[\;q\;∥\;p\;\right]+L\left[\;q\;\right],$$
where
(23)$$\begin{array}{cc} \mathrm{KL}\left[\;q\;∥\;p\;\right]\; & =\int q\left(x_{1:k},{\tilde{R}}_{1:k}\right)\ln\frac{q(x_{1:k},{\tilde{R}}_{1:k})}{p(x_{1:k},{\tilde{R}}_{1:k}\;|\;y_{1:k})} \end{array}$$
(24)$$\begin{array}{cc} L[\;q\;] & =\int q\left(x_{1:k},{\tilde{R}}_{1:k}\right)\ln\frac{p(x_{1:k},{\tilde{R}}_{1:k},\;y_{1:k})}{q(x_{1:k},{\tilde{R}}_{1:k})}. \end{array}$$*p* is the true distribution that is intractable for non-Gaussian noise models, and *q* is a tractable approximate distribution.

In probability theory, a measure of the difference between two probability distributions *p* and *q* is the Kullback–Leibler divergence, denoted as KL[·]. If we allow any possible choice for *q* such as the Gaussian distribution, then lower bound L[q] is maximum when the KL divergence vanishes; that is, $q\left(x_{1:k},{\tilde{R}}_{1:k}\right)=p\left(x_{1:k},{\tilde{R}}_{1:k}\;|\;y_{1:k}\right)$. To minimize the KL divergence, we seek the member of a restricted family of $q(x_{1:k},{\tilde{R}}_{1:k})$. Indeed, maximizing L[q] is equivalent to minimizing another new KL divergence [52], and thus the minimum occurs when factorized distributions $q\left(x_{1:k},{\tilde{R}}_{1:k}\right)=q\left(x_{1:k}\right)\;q\left({\tilde{R}}_{1:k}\right)$ and the following Equations (Equation 25) and (Equation 26) hold,
(25)$$\begin{array}{cc} \ln\;q(x_{1:k}) & =\ln\;p\left(x_{1}\right)+\sum_{t=2}^{k}E_{q({\tilde{R}}_{1:t})}\left[\ln\;p\left(x_{t}\;|\;x_{t-1}\right)\right]+\sum_{t=1}^{k}E_{q({\tilde{R}}_{1:t})}\left[\ln\;p\left(y_{t}\;|\;x_{t},{\tilde{R}}_{t}\right)\right]+\cdots \end{array}$$
(26)$$\begin{array}{cc} \ln\;q({\tilde{R}}_{k}) & =E_{q(x_{1:k})}\left[\;\ln\;p\left(y_{k}\;|\;x_{k},\;{\tilde{R}}_{k}\right)\;\right]+\ln\;p\left({\tilde{R}}_{k}\right)+\cdots , \end{array}$$
where $E_{q(\cdot )}$ represents the expectation with respect to q(·). With assuming that initial state $x_{1}$ is Gaussian, the measurement update with varying noise covariance $E\left[\;{\tilde{R}}_{t}^{-1}\;\right]=\Lambda_{t}^{-1}$, which closely resemble the EKF updates, solve Equation (Equation 25). Algorithm 2 describes the details of the updates.

Now let us assume that the true priors are IID noise models as the case in this study; that is, $p({\tilde{R}}_{k})$ follows $W^{-1}\left(\nu R,\;\nu \right)$ distribution. Then, second term $\ln p({\tilde{R}}_{k})$ in the right-hand side of Equation (Equation 26) is computed using the pdf of the inverse-Wishart distribution in Equation (Equation 21) with its prior noise model.
(27)$$\ln p\left({\tilde{R}}_{k}\right)=-\frac{\nu +m+1}{2}\ln\left|{\tilde{R}}_{k}\right|\;-\frac{\nu }{2}\;\mathrm{tr}\left(R\;{\tilde{R}}_{k}^{-1}\right).$$

As the term is conjugate prior for Equation (Equation 20), the approximations of $q({\tilde{R}}_{k})$ have same mathematical forms as priors; that is, $q({\tilde{R}}_{k})$ also follows $W^{-1}\left({\tilde{\nu }}_{k}\;\Lambda_{k},\;{\tilde{\nu }}_{k}\right)$ distribution.
(28)$$\ln q\left({\tilde{R}}_{k}\right)=-\frac{{\tilde{\nu }}_{k}+m+1}{2}\ln\left|{\tilde{R}}_{k}\right|-\frac{{\tilde{\nu }}_{k}}{2}\;\mathrm{tr}\left(\Lambda_{k}\;{\tilde{R}}_{k}^{-1}\right).$$As $y_{t}\;|\left\{x_{t},{\tilde{R}}_{t}\right\}∼N\left(h\left(x_{t}\right),{\tilde{R}}_{t}\right)$,
(29)$$E\left[\;\ln p(y_{k}\;|\;x_{k},{\tilde{R}}_{k})\;\right]=-\frac{1}{2}\ln\left|{\tilde{R}}_{k}\right|-\frac{1}{2}\;\mathrm{tr}\left(E\left[\zeta_{k}\;\zeta_{k}^{T}\right]\;{\tilde{R}}_{k}^{-1}\right).$$From Equations (Equation 26)–(Equation 29), to handle measurement outliers, similar to Agamennoni et al. [18,19]’s derivation, we derive how to compute precision matrix $\Lambda_{k}$ of approximate distribution $q({\tilde{R}}_{k})$ of ${\tilde{R}}_{k}$ as follows,
(30)$$\begin{array}{cc} {\tilde{\nu }}_{k} & =1+\nu ,\;\;\;{\tilde{\nu }}_{k}\;\Lambda_{k}=E\left[\zeta_{k}\;\zeta_{k}^{T}\right]+\nu R\\ \Rightarrow \Lambda_{k} & =\frac{\nu R+E[\zeta_{k}\;\zeta_{k}^{T}]}{\nu +1}, \end{array}$$
where each feature from one image is independent and
$$\begin{array}{cc} {\left(\zeta_{j}\right)}_{k} & ={\left(y_{j}\right)}_{k}-h\left(\;x\left(t_{\mathrm{img}}\right)\;\right)\\ & ={\left(y_{j}\right)}_{k}-h\left(\;{\left({\hat{x}}_{k}\right)}_{j}+e_{j}\;\right)\\ & \approx {\left(y_{j}\right)}_{k}-h\left(\;{\left({\hat{x}}_{k}\right)}_{j}\;\right)-C_{j}\;e_{j}. \end{array}$$

Next, in Equation (Equation 30),
(31)$$\begin{array}{cc} E\left[\;{\left(\zeta_{j}\right)}_{k}\;{\left(\zeta_{j}\right)}_{k}^{T}\;\right]\; & =\left[{\left(y_{j}\right)}_{k}-h\left(\;{\left({\hat{x}}_{k}\right)}_{j}\;\right)\right]{\left[{\left(y_{j}\right)}_{k}-h\left(\;{\left({\hat{x}}_{k}\right)}_{j}\;\right)\right]}^{T}\ldots \\ \; & \;\;\;+C_{j}\;E\left[e_{j}\;e_{j}^{T}\right]\;C_{j}^{T}\\ \; & =r_{j}\;r_{j}^{T}+C_{j}\;{\left(P_{k}\right)}_{j}\;C_{j}^{T}, \end{array}$$
where estimation error $e_{j}=x_{k}-{\left({\hat{x}}_{k}\right)}_{j}$ and the Jacobian $C_{j}=\frac{\partial h_{j}}{\partial x}{|}_{{\left({\hat{x}}_{k}\right)}_{j}}$. In the sequential measurement update, ${\left({\hat{x}}_{k}\right)}_{j}$ and ${\left(P_{k}\right)}_{j}$ are corrected by Kalman gain $K_{j}$ that is a function of ${\left(\Lambda_{j}\right)}_{k}$, so these update steps are coupled. Hence no a closed-form solution exists, and we can only solve iteratively. The purpose of the iteration seems to be similar to that of the online learning of unknown variances of each noise [10]. In addition, similar to Agamennoni et al.’s interpretation [19], the convergence and optimality of the derived update steps for outliers are guaranteed since the variational lower bound is convex with respect to ${\left({\hat{x}}_{k}\right)}_{j}$, ${\left(P_{k}\right)}_{j}$, and ${\left(\Lambda_{j}\right)}_{k}$. In particular, as the *j*-th feature is observed countless times (i.e., $\nu_{j}$→*∞*), $\Lambda_{j}$ converges to *R* in the limit of an infinitely precise noise distribution, so the iterative update steps reduce to the standard sequential measurement update of the EKF.

If true state $x(t_{\mathrm{img}})$ is significantly different from its estimate ${\left({\hat{x}}_{k}\right)}_{j}$, then statistics $E\left[\;{\left(\zeta_{j}\right)}_{k}\;{\left(\zeta_{j}\right)}_{k}^{T}\;\right]$ dominates, and ${\left(\Lambda_{j}\right)}_{k}$ becomes much larger than *R*. This $\zeta_{j}$ is regarded as a measurement outlier at time *k*. As Kalman gain $K_{j}$ is a function of the inverse of precision matrix ${\left(\Lambda_{j}\right)}_{k}$, the larger ${\left(\Lambda_{j}\right)}_{k}$ values, the smaller the Kalman gain. Therefore, to deal with situations where measurement outliers occur, the iteration for Equations (Equation 30) and (Equation 31) corrects the state estimates and its covariance with low weights.

## 5. Implementation

### 5.1. Marginalization of Feature States

If measurement outliers often occur, a few numbers of sequential updates in the EKF are proceeded to correct the state estimates. Without a sufficient number of measurement updates, the EKF is not robust and even diverges. Hence, outlier-adaptive filtering introduced in Section 4.3 performs the modified measurement update even when a residual is detected as an outlier. Indeed, to save computation resources, this study operates the outlier-adaptive filtering for only features detected frequently outliers. For implementation, we count how many numbers of features augmented in state variables are detected as outliers. Once updating feature outliers by the outlier-adaptive filtering approach, we prune the used feature states from the state vector (Figure 3). In addition, similar to that mentioned in Appendix B, to maintain a certain size of the state vector, after the feature initialization, we marginalize the features with the least number of observations among tracked features.

### 5.2. Summarized Algorithm

This section summarizes and describes an implementation of the proposed method. Figure 4 and Algorithm 2 illustrate a flow chart and the pseudocode of the overall process of the outlier-adaptive filtering approach for V-INS, respectively. For the robust outlier-adaptive filter presented in this paper, the blue boxes in Figure 4 are extended from the figure in [25].
**Algorithm 2** The Outlier-Adaptive Filtering
Require: ${\hat{x}}_{0}^{+}\left(={\hat{x}}_{V_{0}}^{+}\right),P_{0}^{+},Q,R,\;\chi^{2}$1:**for**k=1:T**do**2:                ▹ Image processing front-end in different thread3: **if** new image capture **then**4:*Image Processing*:                       ▹ Algorithm 15:  Stereo matching between current images of left camera $c_{1}$ and right camera $c_{2}$6:  RANSAC between previous and current images of camera $c_{1}$7:  RANSAC between previous and current images of camera $c_{2}$8: **end if**9:                   ▹ Filtering back-end in different thread10: **if** new IMU packet arrival **then**11:*Time Update*:12:  State prediction13: **end if**14: **if** new vision data packet arrival **then**15:  **for**
j=1:♯ of observed features *N*
**do**16:   **if** new feature **then**17:*Feature Initialization*:18:    If any depth of $c_{1}$ or $c_{2}$ is negative, *j*-th feature is outlier19:   **else**                        ▹ tracked feature20:*Outlier Gating Test*:21:    $r_{j}=y_{j}-h_{j}\left({\left({\hat{x}}_{k}\right)}_{j-1}\right)$22:    $S_{j}={\left(C_{j}\right)}_{k}\;{\left(P_{k}\right)}_{j-1}\;{\left(C_{j}\right)}_{k}^{T}+R$         ▹${\left(C_{j}\right)}_{k}=\frac{\partial h_{j}}{\partial x}{|}_{{\left({\hat{x}}_{k}\right)}_{j-1}}$23:    $\gamma =r_{j}^{T}S_{j}^{-1}r_{j}\overset{?}{<}\chi_{j}^{2}$24:*Measurement Updates*:25:    **if** outlier detected **then**26:     ${\tilde{x}}_{j}\leftarrow {\left({\hat{x}}_{k}\right)}_{j-1},\;\;\;{\tilde{P}}_{j}\leftarrow {\left(P_{k}\right)}_{j-1}$    ▹${\left({\hat{x}}_{k}\right)}_{0}={\hat{x}}_{k}^{-},\;{\left(P_{k}\right)}_{0}=P_{k}^{-}$27:     **while** until converged **do**28:      Update measurement noise given the state29:      ${\tilde{r}}_{j}=y_{j}-h_{j}\left({\tilde{x}}_{j}\right)$30:      ${\tilde{C}}_{j}=\frac{\partial h_{j}}{\partial x}{|}_{{\tilde{x}}_{j}}$31:      $W_{j}={\tilde{r}}_{j}\;{\tilde{r}}_{j}^{T}+{\tilde{C}}_{j}\;{\tilde{P}}_{j}\;{\tilde{C}}_{j}^{T}$32:      $\Lambda_{j}\leftarrow \frac{\nu_{j}\;R+W_{j}}{\nu_{j}+1}$33:      Update the posteriori state given noise34:      ${\tilde{S}}_{j}={\left(C_{j}\right)}_{k}\;{\left(P_{k}\right)}_{j-1}\;{\left(C_{j}\right)}_{k}^{T}+\Lambda_{j}$35:      ${\tilde{K}}_{j}={\left(P_{k}\right)}_{j-1}\;{\left(C_{j}\right)}_{k}^{T}\;{\tilde{S}}_{j}^{-1}$36:      ${\tilde{x}}_{j}\leftarrow {\left({\hat{x}}_{k}\right)}_{j-1}+{\tilde{K}}_{j}\;r_{j}$37:      ${\tilde{P}}_{j}\leftarrow {\left(P_{k}\right)}_{j-1}-{\tilde{K}}_{j}\;{\left(C_{j}\right)}_{k}\;{\left(P_{k}\right)}_{j-1}$38:     **end while**39:     ${\left({\hat{x}}_{k}\right)}_{j}={\tilde{x}}_{j},\;\;\;{\left(P_{k}\right)}_{j}={\tilde{P}}_{j}$40:    **else**             ▹ standard sequential EKF in Section 241:     $K_{j}={\left(P_{k}\right)}_{j-1}\;{\left(C_{j}\right)}_{k}^{T}\;S_{j}^{-1}$42:     ${\left({\hat{x}}_{k}\right)}_{j}={\left({\hat{x}}_{k}\right)}_{j-1}+K_{j}\;r_{j}$43:     ${\left(P_{k}\right)}_{j}={\left(P_{k}\right)}_{j-1}-K_{j}\;{\left(C_{j}\right)}_{k}\;{\left(P_{k}\right)}_{j-1}$44:    **end if**45:   **end if**46:  **end for**                 ▹${\hat{x}}_{k}^{+}={\left({\hat{x}}_{k}\right)}_{N},\;P_{k}^{+}={\left(P_{k}\right)}_{N}$47: **end if**48:**end for**

## 6. Flight Datasets Test Results

To examine the influence of outliers in V-INS and validate the reliability and robustness of the proposed outlier-adaptive approach for navigation systems with outliers, we test one of benchmark flight datasets, the so-called “EuRoC MAV datasets” [53]. The visual-inertial sequences of the datasets were recorded onboard a micro aerial vehicle (MAV) while a pilot manually flew around the indoor motion capture environment. For more details, see Appendix C. To articulate the significance of outliers, we select two datasets of the bright scene, called “EuRoC V1 Easy”, and motion blur, called “EuRoC V1 Difficult.” As the images in the difficult dataset are dark scene or motion blur, we hypothesize that outliers occur more frequently in the difficult dataset.

The EKF estimates the relative location and orientation from a starting point. As we do not know the exact absolute location of the origin of given datasets, to compare with ground-truth data given in the datasets, we require certain evaluation error metrics such as so-call “absolute trajectory error [54]”. For more details, see Appendix D. The absolute trajectory error as an evaluation error metric yields the following various comparison plots. Figure 5 illustrates the top-down view of the estimated flight trajectory of the difficult dataset.

Figure 6 depict the estimated *x*, *y*, *z* position and their respective estimation errors. All the estimation errors are bounded within each standard deviation σ envelope, so the proposed approach is reliable vision-aided inertial navigation under even poor illumination environments. Conceptually, the position error gets locked in, and it tends to increase with the length of the trajectory until new features are being mapped.

Similar to the analysis presented in [25], the adaptive filter is a well-tuned estimator since the performance of doing runs with ×3 or ×10 ( /3 or /10) multiplier on the *R* matrix used in the filter is worse for all of those, shown in Table 1. That is, the fact that using the multipliers reveals larger root mean square (RMS) estimation errors indicates that our filter is well-tuned.

Figure 7 shows the advantages of the addition of the outlier adaptation proposed in this paper by comparing it with a baseline, the standard EKF in V-INS.

As Lee [25] and other researchers showed that the standard EKF is a basic filter in V-INS, we choose the method as a baseline here. The baseline only rejects outliers whenever the chi-squared test fails, whereas the outlier-adaptive filtering follows all proposed adaptive approaches in Algorithm 2. Although the iteration in the outlier-adaptive filtering might increase computational resources, it significantly improves the accuracy of estimation. Fortunately, the “while” loop iteration in Algorithm 2 rapidly converges to the optimal noise covariance by twice or three times iterations. For sensitivity analysis, RMS position errors resulting from the baseline and the outlier-adaptive filtering are compiled in Table 2.

Motion blur datasets are more sensitive to outliers as the improvement is larger when applied to those datasets. Although the outlier-adaptive filtering is the best choice for motion blur datasets, we can select an adequate mode depending on computation margin and cost.

Although a number of researchers have investigated V-INS of the EuRoC datasets [55], only a few of them thoroughly has focused on vision measurements with outliers. Table 3 reveals that the proposed estimator, the outlier-adaptive filter, outperforms other state-of-the-art V-INS techniques, called “SVO+MSF [56,57]” and “S-MSCKF [48,58] ” in which stereo is available. As SVO+MSF is loosely coupled, its algorithm actually gets diverged.

## 7. Discussion

This paper has presented practical outlier-adaptive filtering for a vision-aided inertial navigation systems (V-INS) and evaluated its performance with flight datasets testing. In other words, this study develops a robust and adaptive state estimation framework for V-INS under frequent outliers occurrence. In the image processing front-end of the framework, we propose the improved utilization of outlier removal techniques. In filtering back-end, for estimating the states of V-INS with measurement outliers, we implement a novel approach of the outlier-robust extended Kalman filter (EKF) to V-INS, for which we derive iterative update steps for computing the precision noise matrices of vision outliers when the Mahalanobis gating test detects remaining outliers.

To validate the accuracy of the proposed approach and compare it with other state-of-the-art V-INS algorithms, we test the performance of V-INS employing the outlier-adaptive filtering algorithm in the realistic benchmark flight datasets. In particular, to show more improvements of our method over the others’ approaches, we use the fast motion and motion blur flight datasets. Results from the flight datasets testing show that the novel navigation approach in this study improves the accuracy and reliability of state estimation in V-INS with frequent outliers. Using the outlier-adaptive filtering reduces the root mean square (RMS) error of the estimates and accelerates the robustness of the estimates, especially for the motion blur datasets.

The primary goals of future work are listed as follows. Since an inertial measurement unit (IMU) is also a sensor, it could generate outliers in V-INS. With accounting for the process outliers, the accuracy and robustness of the estimator would be improved. If we distinguish process outliers from IMU sensors with measurement outliers from vision data, the extended outlier-robust EKF [32] may be an impressive and innovate approach for this case. Furthermore, the investigation of color noise in V-INS is another possible future work. One of the required assumptions of the Kalman filter is the whiteness of measurement noise. As an illustration, during sampling and transmission in image processing, color noise that may be originated from a multiplicity of sources could degrade the quality of images [59]. The vibrational effects of camera sensors might also produce colored measurement noise [60]. That is, if the residuals of vision data are correlated with themselves at different timestamp, then colored measurement noise occurs in V-INS. Therefore, the images with color noise would be filtered for ensuring the accuracy of locating landmarks. As modeling noise without additional prior knowledge of the noise statistics is typically difficult, the machine learning techniques-based state estimator for colored noise [61,62] may handle the unknown correlations in V-INS.

This study showcases the approaches using stereo cameras but is also suitable for monocular V-INS and employable to other filter-based V-INS frameworks. We test benchmark flight datasets to validate the reliability and robustness of this study, but additional validating with other flight datasets or real-time flight tests would help to prove its more robustness. In addition, we can operate unmanned aerial vehicles (UAVs) stacked the navigation algorithms in this study and a controller-in-the-loop. The use of the controller-in-the-loop could be a more important validation criteria due to the potential for navigation-controller coupling. The research objectives and contributions presented here will remarkably advance the state-of-the-art techniques of vision-aided inertial navigation for UAVs.

## Figures and Tables

**Figure 1 sensors-20-02036-f001:**
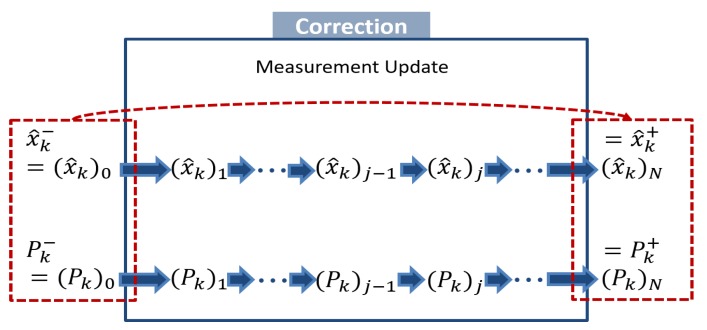
A schematic of the sequential measurement update.

**Figure 2 sensors-20-02036-f002:**
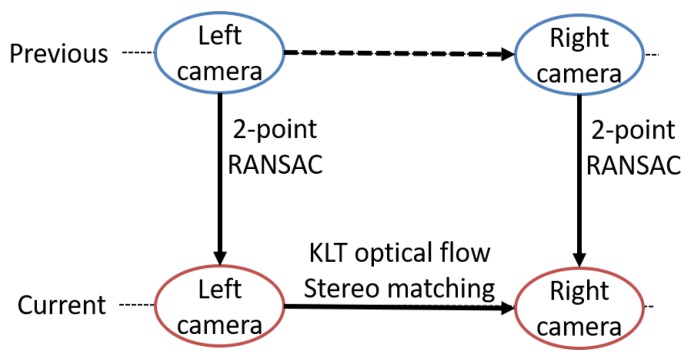
Close loop steps of outlier rejection in image processing front-end.

**Figure 3 sensors-20-02036-f003:**
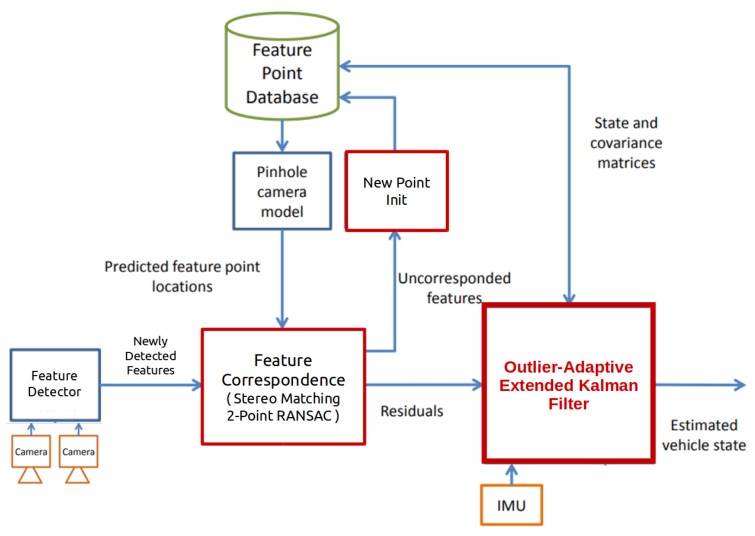
A block diagram of the vision-aided inertial navigation system employing the outlier-adaptive filtering.

**Figure 4 sensors-20-02036-f004:**
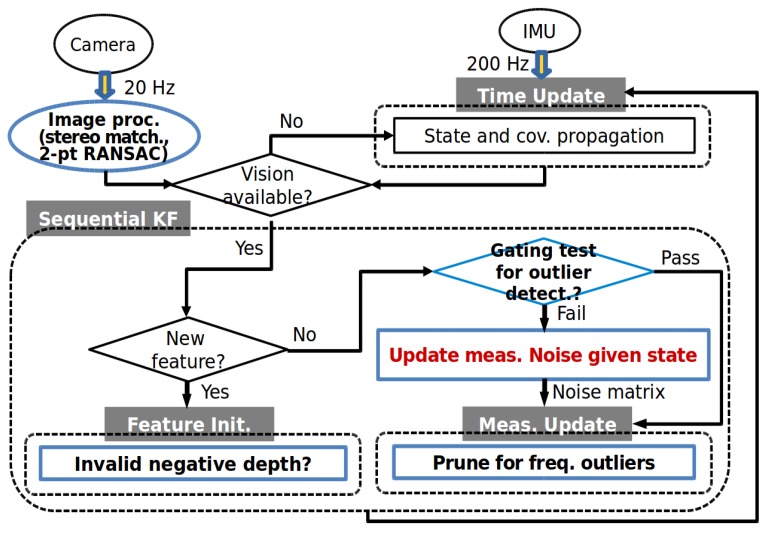
A flow chart of the overall process of the outlier-adaptive filtering.

**Figure 5 sensors-20-02036-f005:**
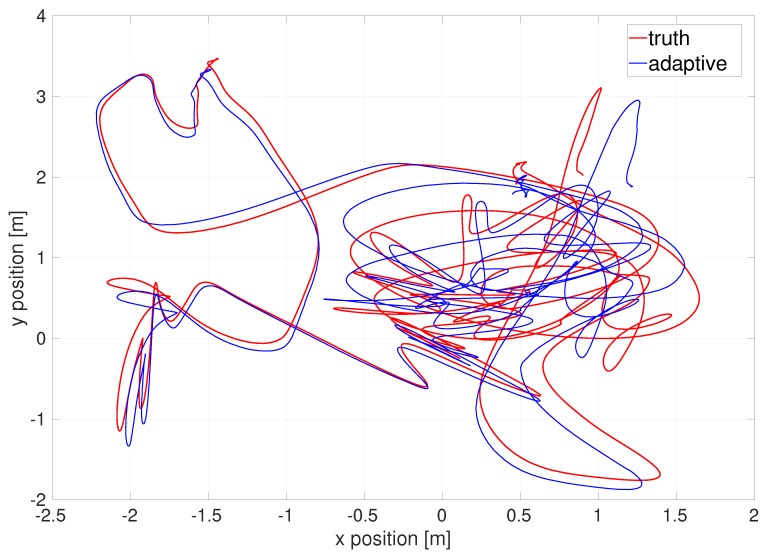
Top-down view of flight trajectory of the EuRoC V1 difficult dataset by the outlier-adaptive filter.

**Figure 6 sensors-20-02036-f006:**
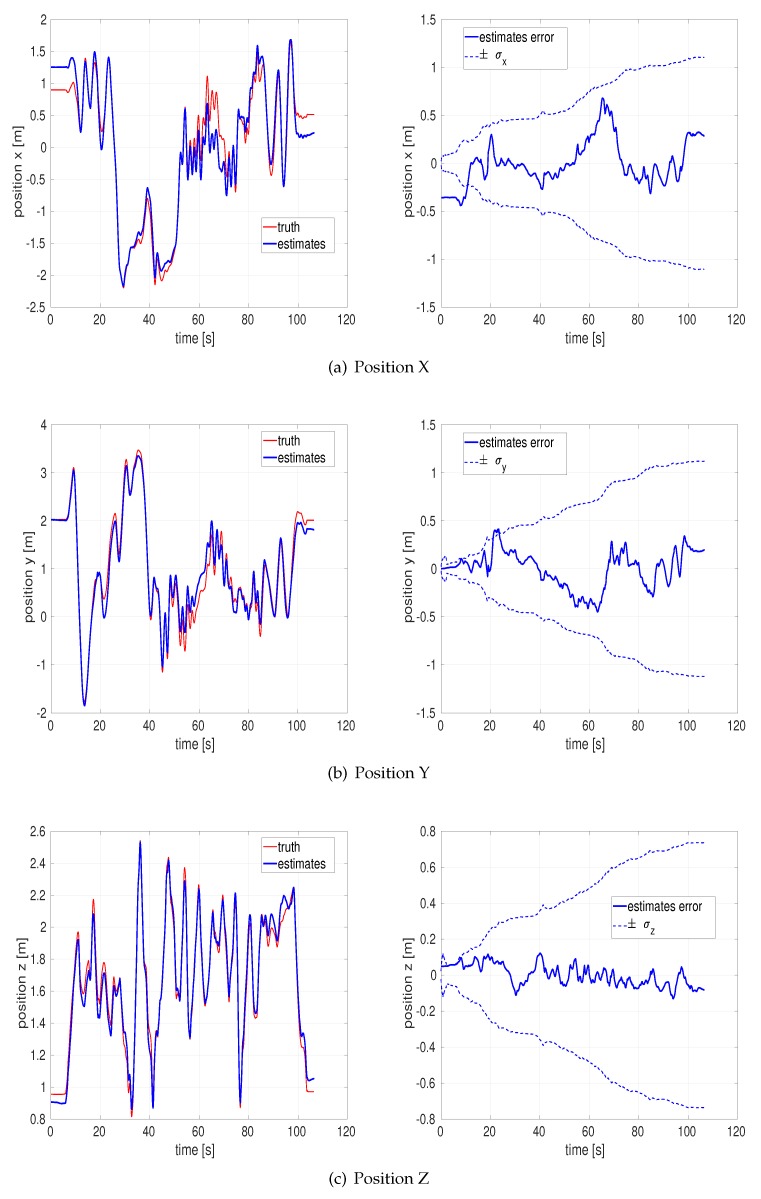
Position and estimation error of the EuRoC V1 difficult dataset by the outlier-adaptive filter.

**Figure 7 sensors-20-02036-f007:**
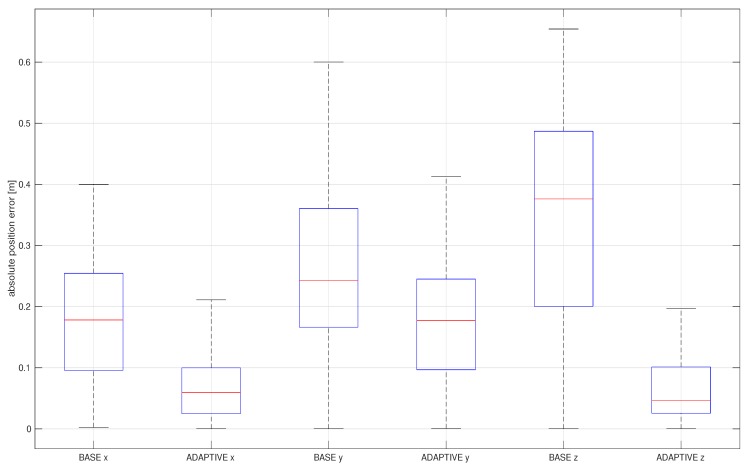
Box plot of absolute estimation error of the position of the EuRoC V1 difficult dataset by the outlier-adaptive filter.

**Table 1 sensors-20-02036-t001:** Indication that the outlier-adaptive filter is well-tuned for the EuRoC V1 difficult dataset.

Multiplier on *R*		/10	/3	1	×3	×10
RMS error [m]		0.9240	0.3801	0.1700	0.5153	0.5610

**Table 2 sensors-20-02036-t002:** Sensitivity analysis in RMS position error [m] of the outlier-adaptive filtering.

Dataset		EuRoC V1 Easy		EuRoC V1 Difficult
		Slow Motion 0.41 m/s, 16.0 deg/s		Fast Motion 0.75 m/s, 35.5 deg/s
Method		*Bright Scene*		*Motion Blur*
Baseline		0.2558		0.3656
Outlier-Adaptive		0.2237		0.2264

**Table 3 sensors-20-02036-t003:** Comparison with other methods in RMS position error [m] of the outlier-adaptive filtering.

Dataset		EuRoC V1 Easy	EuRoC V1 Difficult
Method		*Bright Scene*	*Motion Blur*
Outlier-Adaptive Filter		0.2237	0.2264
SVO+MSF [56,57] (loosely coupled)		0.40	×
S-MSCKF [48,58] (stereo-filter)		0.34	0.67

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
