# Peer review of "Robust Outlier-Adaptive Filtering for Vision-Aided Inertial Navigation"

_sensors, 2020, doi:10.3390/s20072036_

Round 1

Reviewer 1 Report

Well written. I enjoyed reading it. Thank you.

Author Response

Thank you for the review.

Reviewer 2 Report

This paper has presented a outlier-adaptive filtering for vision-aided inertial navigation systems. But in fact, this method just simply combines the Mahalanobis gating test with the ORKF. In the whole exposition, I didn't see a good innovation point. This paper introduced in detail the existing methods used in the author's V-INS system, but the introduction of their own method is very few, so I don't see the theory of the method proposed in this paper and the advantages of this method. In addition, the introduction part of the paper lacks the introduction of the current methods of eliminating outliers of V-INS. In the experimental part, there is no comparison between the proposed method and other methods. Thus, I think it is necessary to clearly point out the innovation points and describe the connotation of innovation points.

Reviewer 3 Report

  1. For the test, it is better to make clear statement of the sensors used, reference accuracy, etc. for readers who are not familiar with the datasets.
  2. For the inertial sensor biases, random constant is applied in Eq (13) and (14), which is not the case for most IMUS. Please make clear statement about what IMU is applied and why random constant is used.
  3. In the test part, the origin is unknown, which seems to be conflict to that the INS mechanization and time update in the filter are conducted in the inertial frame. Please make clear statement about what is neglected. 

Round 2

Reviewer 2 Report

I have read the author's reply and agree with the author's description of the innovation of this article.